# One of these things is not like the other: Mixed predator cues result in lopsided phenotypic responses in a Neotropical tadpole

**Dean M. Rosenthal, Luana Deng, Tarif Rose, Justin C. Touchon** *

Biology Department, Vassar College, Poughkeepsie, New York, United States of America

* jutouchon@vassar.edu

**Data Availability Statement:** The data used in this paper are archived at Data Dryad and can be accessed at https://doi.org/10.5061/dryad.1g1jwsv1g.

## Abstract

Many organisms have evolved to produce different phenotypes in response to environmental variation. *Dendropsophus ebraccatus* tadpoles develop opposing shifts in morphology and coloration when they are exposed to invertebrate vs vertebrate predators. Each of these alternate phenotypes are adaptive, conferring a survival advantage against the predator with which tadpoles were reared but imposing a survival cost with the mismatched predator. Here, we measured the phenotypic response of tadpoles to graded cues and mixed cues of both fish and dragonfly nymphs. Prey species like *D. ebraccatus* commonly co-occur with both of these types of predators, amongst many others as well. In our first experiment, tadpoles increased investment in defensive phenotypes in response to increasing concentrations of predator cues. Whereas morphology only differed in the strongest predation cue, tail spot coloration differed even at the lowest cue concentration. In our second experiment, tadpoles reared with cues from both predators developed an intermediate yet skewed phenotype that was most similar to the fish-induced phenotype. Previous studies have shown that fish are more lethal than dragonfly larvae; thus tadpoles responded most strongly to the more dangerous predator, even though the number of prey consumed by each predator was the same. This may be due to *D. ebraccatus* having evolved a stronger response to fish or because fish produce more kairomones than do dragonflies for a given amount of food. We demonstrate that not only do tadpoles assess predation risk via the concentration of predation cues in the water, they produce a stronger response to a more lethal predator even when the strength of cues is presumed to be identical.

## Introduction

In the face of environmental variation, organisms must adapt in order to survive. While natural selection mediates genetic change in populations over time, phenotypic plasticity is the capability of an organism's single genotype to produce multiple phenotypes in response to varying environmental conditions [1, 2]. One such example is predator-induced plasticity, a phenomena in which prey develop adaptive, predator-specific phenotypes which decrease risk of capture [1, 3, 4].

**Funding:** The author(s) received no specific funding for this work.

**Competing interests:** The authors have declared that no competing interests exist.

In order to develop predator-induced phenotypes, organisms must first detect predation risk in the environment. Adaptive phenotypes are generally induced via detection of passive biological or environmental features which may provide relevant information about nearby risks [5]. Although multiple sensory modalities are likely involved in predator-cue detection [6], aquatic chemicals are often inadvertently released by predators when they consume prey and are widely used by potential prey co-habiting the same environment [7, 8]. Depending on the predator-prey relationship, the phenotype-inducing cue may be a combination of kairo-mones released from the predator alone, alarm pheromones produced by conspecific prey under attack, density based cues of conspecifics, and/or compounds and tissues derived from the mastication and digestion that occur with predation events [9–12].

Predator-induced plasticity has been thoroughly studied in aquatic zooplankton and gastro-pods [13, 14]. For example, in response to predator cues freshwater cladocerans and rotifers can develop spiny bodies and snails will thicken their shells, both in an attempt to reduce the likelihood of being eaten [15, 16]. Among vertebrates, larval amphibians are perhaps the best models for studying plasticity due to their rapid development and potential for plasticity in behavior, life-history, and morphology. Many tadpole species respond to predation cues by reducing activity levels and altering the timing of metamorphosis [17]. Predation cues may also stimulate plasticity in tail shape, coloration, and body size, though plastic responses vary depending on both amphibian and predator taxa [18–20].

The pantless treefrog (Hylidae; *Dendropsophus ebraccatus*), native to Central and South America, is one species in which tadpoles produce different phenotypes in response to more than one type of predator [19]. Predation cues from aquatic insects (e.g., dragonfly larvae, giant water bugs, etc.) stimulate tadpoles to develop a large, deep red and black tail and cause a reduc-tion in activity [21–23]. Alternatively, predation cues from fish stimulate tadpoles to develop shallower, achromatic tails and cause an increase in activity [21, 22, 24]. Importantly, these adjustments to the phenotype are heritable [25], adaptive, and improve survival with the induc-ing predator but reduce survival with a mismatched predator [21]. Large colorful tails functions as a lure to draw predation strikes towards the tail instead of the body [26, 27]. Furthermore, tail shape greatly influences swimming performance irrespective of predator-induction [28, 29].

The environmental specificity of each phenotype begs several questions. First, can tadpoles accurately assess variation in the concentration of cues indicating potential risk? Second, can tadpoles assess conflicting cues of risk—as likely occurs in nature—and what is the resulting phenotype of tadpoles raised with cues of multiple predators? Both of these questions have been addressed in various invertebrate taxa. For example, *Helisoma* snails that are raised with increasing concentrations of cues from crayfish predators respond by proportionally increas-ing shell thickness, although such a response plateaus eventually [30]. Similarly, *Daphnia pulex* exposed to increasing concentrations of predator kairomones responded with a graded increase in defensive morphology which also plateaued [31]. Considering instead how prey respond to combined cues from different predators, results generally indicate that prey are capable of integrating multiple sources of information and that cues interact to partially negate one another. For example, *Daphnia galeata* will reduce time to maturation in response to each of two predators, but when mixed one cue effectively blocks the detection of the other, leading to a response that is less than what might be predicted based on individual effects alone [32]. Snails exposed to simultaneous cues of two different predators do not fully respond to either predator, but are able to respond to cues at lower concentrations than when exposed to cues from a single predator [30]. Indeed, prey responding to cues of multiple predators often pro-duce phenotypes that are biased towards one particular predator [32, 33].

In vertebrates, various forms of phenotypic plasticity in embryos and tadpoles have been quantified in a large number of species, mostly (but not exclusively) from North America and

Europe [34–36]. However, just two species of Ranid frogs (true frogs) are known to respond to increasing concentrations of predation cues with graded investment in defensive morphologies [37, 38] and three species of Ranids have been exposed to simultaneous cues from multiple predators [39, 40]. To our knowledge tadpoles from other families of frogs, such as the treefrogs (Hylidae) have not yet been tested.

*Dendropsophus ebraccatus* tadpoles certainly face risk from many types of predators in nature. Touchon and Vonesh [41] surveyed five ponds in Panama in two different years and found that multiple species of fish and dragonfly larvae co-occurred on three occasions, not to mention the presence of numerous other predators such as giant water bugs, diving beetle larvae, fishing spiders and freshwater shrimp. Thus, understanding how these species respond to more realistic predation scenarios is vital to understanding how phenotypic plasticity has evolved in the face of conflicting selection pressures present in the natural world. Moreover, the clades containing true frogs and treefrogs likely diverged from one another more than 120 million years ago [42], and as such it is important to increase the breadth of amphibian taxa which have been studied in order to gain a more general understanding of predator-induced plasticity.

We raised *D. ebraccatus* tadpoles with predation cues from fish and dragonfly nymphs and measured phenotypic responses to both graded and mixed cues in two separate experiments. We hypothesized that tadpoles would respond to increasing concentrations of predation cues by producing concomitantly more extreme phenotypes. Similarly, we predicted that exposure to mixed cues would produce intermediate phenotypes between that of tadpoles raised with dragonfly or fish cues alone, but that this intermediate phenotype would be more similar to a fish-induced phenotype than a dragonfly-induced phenotype. This prediction is based on the fact that *D. ebraccatus* tadpoles face a higher risk of predation from fish than from dragonfly nymphs [21, 41].

## Materials and methods

### Breeding

This research was conducted under approved Vassar College IACUC protocols #14-22B and 18-12B. The first experiment was conducted during July and August 2017. The second experiment was conducted between September and November of 2019. All research used a colony of *D. ebraccatus* housed at Vassar College in Poughkeepsie, New York. Frogs in the colony were second and third generation offspring of frogs originally from a population in Gamboa, Panama. For both experiments frogs were bred by placing adult males and females in a $50 \times 50 \times 90$ cm rain chamber containing artificial pond water (reverse osmosis water plus Kent Marine R/O Right and Kent Marine pH Stable) and an assortment of bromeliad and philodendron plants. Frogs were "rained on" for at least six hours to stimulate amplexus at which point mating pairs were moved to inflated one gallon zip lock bags with approximately 100 ml of water for egg deposition. The following morning, eggs were gently transferred with forceps from the inner wall of the plastic bag into plastic cups where they were misted with artificial pond water four times per day. Three days into development, cups were flooded with artificial pond water to induce hypoxic stress in embryos and thus stimulate hatching. Tadpoles were allowed to further develop for another two days post-hatching to ensure sufficient development before the start of experiments.

Experiment one used eight unique families, whereas experiment two used six families. After hatching, tadpoles were pooled by family and then haphazardly divided into treatment groups. In experiment one, eight replicates of the experiment were setup (i.e., each family was a complete replicate of all treatments). We used 80 tadpoles from each family, set up in eight groups of 10 tadpoles each in 2 L of water. In experiment two, the experiment was once again

replicated by family (i.e., all treatments were replicated six times). We used 76–88 tadpoles from each family, set up in four sets of approximately 20 tadpoles each in 4 L of water. Extra hatchlings were used to feed predators in order to generate cues for the experiments.

## Rearing tadpoles with predator cues

Experiment one contained four predator cue treatments (high, medium, low and no predator cues) crossed with two predators (mosquitofish, *Gambusia affinis*, or locally caught aeshnid dragonfly larvae) for a total of eight treatments. There were no-predator treatments for each predator type, and they were identical. Predators were maintained alone in 2 L of water and fed four *D. ebraccatus* tadpoles per day, thereby creating a cue concentration of two tadpoles consumed per L of water per day. Closely related species of predators (i.e., in the same genus) co-exist with *D. ebraccatus* in nature and tadpoles in our research colony have previously been shown to respond to locally collected predators [21]. Experimental tadpoles were housed separate from predators in 2 L plastic aquaria (19 cm X 10 cm X 10 cm) containing artificial pond water. 500 ml of water was removed and replaced from each experimental tadpole container each day. The high cue treatment received 500 ml of full-strength cue water, resulting in an effective concentration of 0.5 tadpoles consumed per L per day. Medium and low cue treatments received 200 ml of cue water + 300 ml of artificial pond water or 50 ml of cue water + 450 ml of artificial pond water, resulting in effective concentrations of 0.2 and 0.05 tadpoles consumed per L per day, respectively. The no predator cue treatment received 500 ml of artificial pond water that lacked predation cues. The initial size of tadpoles was 7.3 ± 0.3 mm (total length mean ± SD, here and throughout). Fish were 43.0 ± 2.6 mm and dragonfly larvae were 34.5 ± 2.9 mm. Since tadpoles were housed separately from predators, tadpoles only ever experienced predators via the water that was introduced into their tanks each day.

Experiment two contained four treatments: no-predator control, fish, dragonfly, and combined cues. Tadpoles were placed in the central compartment of a 25 cm × 15 cm × 15 cm 4L plastic aquarium divided into thirds with mesh screens. Two predators were introduced to the lateral compartments of each tank, one on each side: two *Gambusia affinis* for the fish treatment (FF), two locally captured libellulid dragonfly nymphs for the dragonfly treatment (DD), or one of each predator for the combined cue treatment (DF). The lateral compartments of the predator-free control tanks (CC) were left empty. The initial size of tadpoles was 6.4 ± 0.7 mm. Fish were 28.0 ± 4.0 mm and dragonfly larvae were 20.1 ± 1.8 mm. Each predator was fed one *D. ebraccatus* tadpole per day. By using this double predator design, we ensured that equal quantities of predation cues were introduced into each tank (Relyea, 2003). All predator compartments (including controls) contained a stick, which was used as a perch by dragonfly nymphs.

Tanks in both experiments were housed in a climate-controlled room maintained at 27 ºC and on a 12:12 L:D cycle. In both experiments we fed tadpoles daily with rabbit chow coated in Sera micron (Sera, Heinsberg, Germany) which was available *ad libitum*. Predators were fed feeder tadpoles housed in separate aquaria from the experimental tadpoles. In experiment one, predators ate all their feeder tadpoles every day. In experiment two, both fish and dragonfly nymphs ate 0.9 ± 0.1 tadpoles per day. If a predator failed to eat for two days it was replaced. Waste was removed from tadpole tanks on a daily basis. In experiment two half of the water in each tank was exchanged every other day.

## Measuring phenotypes

Both experiments were run for 10 days, at which point tadpoles were anesthetized in 0.2 g/L neutral buffered tricaine methanesulfonate, MS-222 (Sigma-Aldrich) for a few minutes and

photographed from a lateral perspective with a Nikon D7100 DSLR camera with a Tokina 100 mm macro lens and external flash. Each tadpole was placed on its side between two pins in a shallow pan filled with water alongside a ruler for scale and black and white color standards for color correction. MS-222 is widely used as an anesthetic for amphibians and has a considerably shorter recovery time than other caine-derivative anesthetics [43].

## Data analysis

Data were analyzed in R v.4.1.2 [44]. All photographs were color corrected in Adobe Photoshop using the black and white standards in each image. To assess morphology, we used geometric morphometrics [45, 46]. We digitized 14 landmarks using the StereoMorph package in R [47]. Landmarks were the 1) tip of the snout, 2) dorsal surface of the head above the eye, 3) center of the eye, 4) ventral surface of the head below the eye, 5) vent, 6) ventral edge of the tail muscle at the head, 7) dorsal edge of the tail muscle, 8) tip of the tail, not including the filament, 9–14) the dorsal and ventral margins of the tail fin 25%, 50%, and 75% of the distance between the body and tail fin tip (Fig 1A).

*Dendropsophus ebraccatus* tadpoles generally have a pigmented spot at the tip of the tail (Fig 1A), which we traced using the freehand selection tool in Image J [48]. We calculated the area of the tail spot, taking care to exclude patches absent in pigmentation. Some tadpoles lacked a tail spot entirely, in which case we measured a small patch of the tail muscle. Three coloration measurements were obtained for each tail spot—hue, saturation, and brightness—all measured on a scale of 0–255. Hue is the shade of a color, with lower values specifying longer wavelengths of light and higher values decreasing in their corresponding wavelength. Saturation is the colorfulness of an area, with higher values indicating more intense colors. Brightness specifies the amount of light reflected by an area, with higher values indicating more light reflected. Because tadpoles were photographed on a white background with brightness of 255, higher values indicate more transparent tails and lower values indicate more heavily pigmented tails. Each photograph contained a preassigned random number to ensure that all measurements were conducted blindly.

Geometric morphometric data were analyzed using the geomorph package [49, 50]. Landmark positions were aligned using a generalized Procrustes analysis (GPA) and were analyzed with a principal components analysis (PCA) of the coordinates from the Procrustes fit using the gm.prcomp function in geomorph. We analyzed each of the first two principal components with mixed effects models using the lmer function in the lme4 package [51, 52]. There were no significant treatment effects on further principal components and so we did not consider them. In all mixed models, tadpole family and rearing tank were classified as random effects to account for pseudoreplication of tadpoles raised within common genetic backgrounds and rearing environments. For experiment 1, fixed effects were predator cue concentration (treated as a continuous variable based on the concentration of tadpole consumed per day), predator, and their interaction. We also analyzed experiment 1 data treating predator cue as a categorical variable, which allowed post-hoc tests comparing the predator treatments at each cue level. For experiment 2, the lone fixed effect was predator treatment. The fit of each model was assessed by viewing Q-Q plots. Significance of predictors was determined using likelihood ratio tests of nested models. Post-hoc comparisons were conducted using the emmeans function in the emmeans package [53]. In experiment 1, PC1 and PC2 explained 42% of the variation in tadpole shape, and in experiment 2, these components explained 47% of the variation in tadpole shape.

Because tail spot area (TSA) and the three measures of coloration were correlated with one another (all partial correlations between variables in both experiments $P \leq 0.01$ except TSA

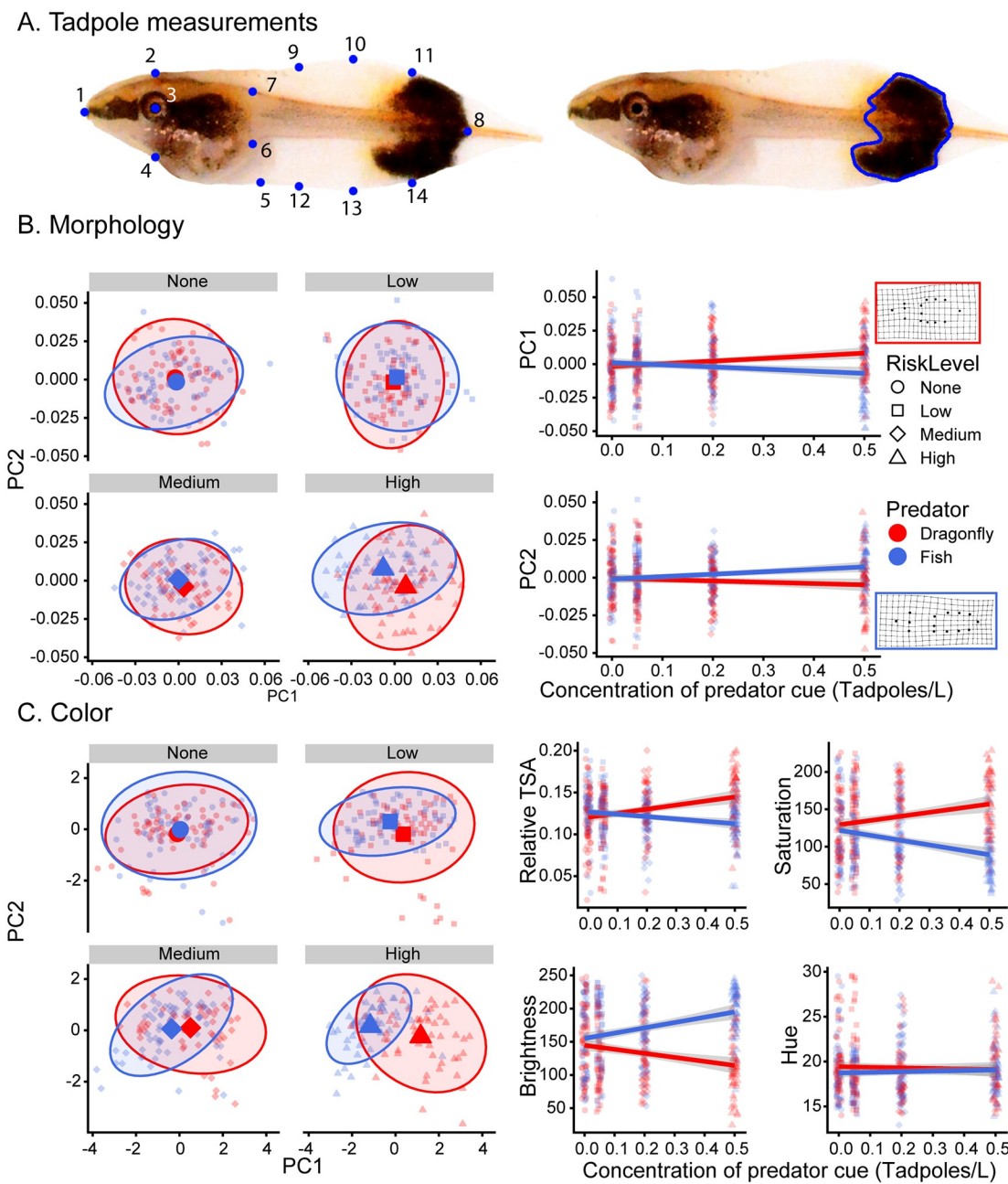

**Fig 1. Plasticity of _Dendropsophus ebraccatus_ tadpoles raised with graded predator cues.** A) Geometric morphometrics utilized 14 landmarks (left) to digitize the size and shape of each tadpole. The tail spot was outlined by hand (right) in order to measure the size and color. B) Morphology was analyzed with a principal components analysis (PCA). The scatterplots (left) show the first two components from the PCA plotted against one another for each of four predator concentrations of fish or dragonfly predator cues (None, Low, Medium or High). Small points represent individual tadpoles, large points represent the mean response for each rearing treatment and shaded regions represent 95% confidence ellipses. Linear regressions show the relationship between PC1 (right, top) and PC2 (right, bottom) and increasing concentrations of fish or dragonfly predator cues. Thin-plate spline deformation grids are inset to show the change of tadpole shape along the main directional axis of change between the two predators, from large PC1 and small PC2 values (red box, dragonfly treatment) to small PC1 and large PC2 values (blue box, fish treatment). C) Tail spot coloration was also analyzed with a PCA. The scatterplots (left) show the first two components plotted against one another for each of the four predator cue concentrations. Linear regressions (right) showing the relationship between increasing concentrations of fish or dragonfly predation cues and relative tail spot area and the saturation, brightness and hue of the tail spot. In all regression plots, points have been jittered slightly to improve legibility.

and hue, which were not significantly correlated), we analyzed tail spot size and color with a PCA using the prcomp function in R. The PCA contained four measurements for each individual: tail spot hue, saturation, brightness, and relative tail spot size, which was calculated as the square root of the tail spot area divided by the centroid size from the GPA fit, which provided a measure of overall body size for each tadpole. Square root transformation of tail spot area greatly improved normality. Values were scaled and centered prior to analysis. The first two components of each PCA were analyzed with mixed effects models as above. In experiment 1, the first two principal components explained 83% of the variation in tail spot coloration, and in experiment 2 explained 87% of variation in tail spot color. For ease of interpretation we have also included the original color and tail spot area data in our figures.

## Results

### Experiment 1: The effect of graded cues on predator-induced phenotypic plasticity

Morphology within each treatment was variable but we detected significant changes in defensive tadpole phenotype in response to gradually increasing cues of risk from fish or dragonfly predators. Larger values of PC1 indicated changes in the relative depth of the tail fin, whereas larger values of PC2 primarily indicated shorter tails (Fig 1B). Tadpole morphology diverged as predicted; stronger predator cue concentrations from dragonfly larvae or fish led to increasingly deeper and shorter tails or shallower and longer tails, respectively (Fig 1B). Analysis of each component separately revealed a significant effect of predator on PC2 ($\chi^2 = 24.4$, $P < 0.001$) as well as a significant interaction between predator cue concentration and predator for both PC1 and PC2 (Fig 1B; PC1: $\chi^2 = 8.1$, $P = 0.004$; PC2: $\chi^2 = 6.3$, $P = 0.012$). Analyzing predator cue concentration as a categorical variable yielded similar results. Post-hoc analyses at each cue concentration revealed that the predator treatments differed significantly from one another at the High cue concentration for both PC1 and PC2 (both $P \leq 0.003$), but not at any lower cue concentrations.

Gradual changes in coloration were more pronounced than for morphology. PC1 indicated changes in the relative tail spot area, saturation and brightness and PC2 indicated variation in hue. For PC1, tail spot saturation and size were tightly related; both increased with stronger concentrations of dragonfly nymph cues but decreased with stronger cues of fish predation, and the opposite was true for brightness (Fig 1C). There was a significant effect of predator type and a significant interaction between predator cue concentration and predator type for PC1 (predator: $\chi^2 = 15.4$, $P < 0.001$, predator*concentration: $\chi^2 = 19.30$, $P < 0.001$). PC2, which indicated variation in hue, varied only by predator treatment (Fig 1C; predator: $\chi^2 = 6.3$, $P = 0.01$). As was true for morphology, analyzing predator cue concentration as a categorical variable yielded nearly identical results. Post-hoc analyses of PC1 and PC2 comparing the two predator treatments at each cue concentration revealed that predator treatments understandably did not differ in the cue-free control (both $P \geq 0.40$). However, PC1 differed significantly by predator treatment at Medium and High cue concentrations (all $\leq 0.03$) and was marginally different at the Low cue concentration ($P = 0.079$), and PC2 differed at both Low and High concentrations (both $P \leq 0.05$).

### Experiment 2: The effect of single or combined predator cues on tadpole phenotypes

Tadpoles in our second experiment were exposed to cues from either dragonfly nymphs or fish alone (treatments DD and FF, respectively), to combined cues from the two predators

(treatment DF), or were raised as predator-free controls (treatment CC). Like in experiment 1, morphology within each treatment was admittedly variable, but *D. ebraccatus* tadpoles once again developed significantly different phenotypes as a result of their rearing treatment. Larger values of PC1 indicated deeper and shorter tail fins and smaller values indicated longer and more slender tails, whereas larger values of PC2 primarily indicated larger tails in general (Fig 2A). Tadpoles in the CC treatment had the longest and most slender tails whereas tadpoles raised with just dragonfly nymph cues (DD) had the largest relative tails (Fig 2A). Analysis of each component separately found significant effects of predator treatment (Fig 2A; PC1: $\chi^2$ = 28.1, $P < 0.001$; PC2: $\chi^2$ = 9.0, $P < 0.029$). Post-hoc analyses revealed that for PC1 the FD and FF treatments were not different ($P = 0.63$), but all other treatments were at least marginally significantly different from one another (DD vs CC: $P = 0.06$, all others $P < 0.05$). For PC2, tadpoles exposed to only dragonfly cues (DD) differed from those exposed to just fish (FF, $P = 0.038$), and no other treatments differed (all $P > 0.18$).

Predator-induced changes in tail spot coloration showed a similar pattern, as expected. Tadpoles raised with mixed cues from both dragonfly nymphs and fish developed tail coloration much more similar to that of tadpoles raised with fish alone, as compared with the bold tail coloration of caused by dragonfly nymph cues. PC1 indicated changes in the relative tail spot area, saturation and brightness and PC2 indicated variation in hue. Like in experiment 1, tail spot saturation and size were highly correlated and increased with larger values of PC1, whereas tail brightness increased with smaller values of PC1. There was a significant effect of predator treatment for PC1 ($\chi^2$ = 42.3, $P < 0.001$) but not PC2 ($\chi^2$ = 3.1, $P = 38$). Tadpoles raised with only dragonfly larvae (DD treatment) had the largest and most saturated tail spots with lowest brightness, followed by tadpoles in the CC treatment, then DF and lastly FF treatments (Fig 2B). Post-hoc analyses of PC1 demonstrated that all treatments differed significantly from one another ($P \leq 0.005$) except the DF and FF treatments ($P = 0.33$).

## Discussion

Examples of predator-induced plasticity are widespread in nature, with many species possessing the capacity to develop multiple forms of predator-specific phenotypes [39, 54, 55]. Such plasticity is generally assumed to improve survival when prey are faced with actual predation risk, and the adaptive nature of induced phenotypes has been repeatedly shown both in tadpoles [20, 21, 28, 56] and other prey animals [4, 55]. However, not all predators are created equal; different aquatic predators have different predation styles and capabilities and may therefore exert different selective pressures on prey animals. Furthermore, defensive phenotypes may reduce risk with one predator while increasing risk from another [16, 20, 21]. Given that prey may cohabitate with multiple predators in nature, it is of interest how prey might respond to cues from multiple predators simultaneously. Similarly, since the makeup of predator communities can vary over space or time [24, 41, 57, 58], for example due to intraguild predation or because of temporal fluctuation in the population dynamics of individual species, it is also of interest to know how prey respond to variation in the magnitude of risk they perceive.

The Neotropical tadpole *D. ebraccatus* develops opposing predator-induced phenotypes when exposed to invertebrate vs vertebrate predators, such as dragonfly larvae and fish [19, 25]. Here, we report two findings which add to our overall understanding of predator-induced phenotypic plasticity. In our first experiment we demonstrated that *D. ebraccatus* tadpoles increased investment in defensive phenotypes relative to the concentration of predation cues in their environment. Tadpole shape significantly differed amongst tadpoles reared with the strongest concentration of predation cues, but tail spot size and coloration differed even in the

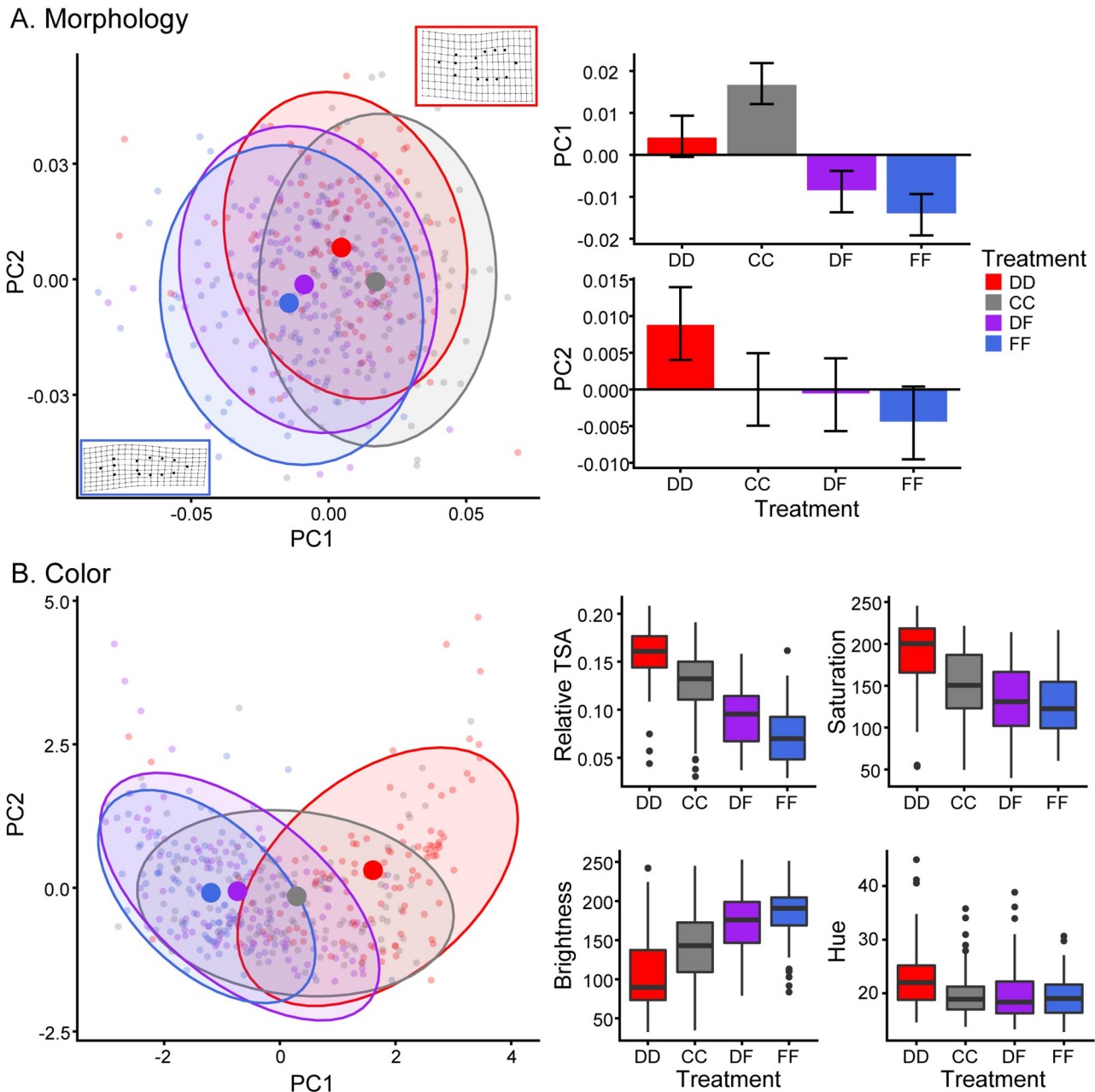

**Fig 2. Plasticity of *Dendropsophus ebraccatus* tadpoles raised with mixed or single predator cues.** A) Morphology was analyzed with a principal components analysis (PCA). The scatterplot (left) shows the first two components from the PCA plotted against one another. Small points represent individual tadpoles, large points represent the mean response for each rearing treatment and shaded regions represent 95% confidence ellipses. Thin-plate spline deformation grids are inset to show the change of tadpole shape along the main directional axis of change between the two predators, from large PC1 and PC2 values (red box, dragonfly treatment) to small PC1 and PC2 values (blue box, fish treatment). Bar graphs show the estimated marginal mean ± standard error for PC1 (right, top) and PC2 (right, bottom). B) Tail spot coloration was also analyzed with a PCA. The scatterplot (left) shows the first two components from the PCA plotted against one another. Points and ellipses are as described above. Box-and-whisker plots (right) show how relative tail spot area and the saturation, brightness and hue of the tail spot varied across the four predator treatments. Plots show the median (thick horizontal line), interquartile range (top and bottom of the colored box) and either the most extreme values (ends of the whiskers) or 1.5 times the interquartile range and outliers (ends of the whiskers followed by points). In all plots, predator treatments are DD: dragonfly cues only, CC: predator-free control, DF: mixed dragonfly and fish cues, and FF: fish cues only.

lowest cue concentration (Fig 1). Thus, the overall defensive phenotype was incrementally strengthened in concert with cues indicating greater risk. In our second experiment, we found that tadpoles reared with cues from both predators developed an intermediate phenotype that was much more similar to the fish-induced phenotype than the dragonfly-induced phenotype. The first component of both the morphology and tail spot coloration PCAs differed significantly by predator treatments, with the exception of the FF and DF treatments, which did not differ from one another (Fig 2). Thus, when considering changes in both shape and tail color, tadpoles exposed to mixed cues of both predators developed a phenotype that was more akin to the phenotype of tadpoles exposed to just fish predation cues, as compared to tadpoles exposed to just dragonfly nymph cues. Collectively, these results demonstrate that prey organisms can produce highly nuanced responses to environmental cues of relative risk.

In our study system, fish are a much more dangerous predator to *D. ebraccatus* tadpoles than are dragonfly nymphs [21, 41]. Not only do fish consume tadpoles more quickly, often engulfing them in a single bite, but they can eat more tadpoles in a given amount of time than can dragonfly larvae. Thus, when facing the threat of two different predators, *D. ebraccatus* produced a response that was biased towards the more dangerous predator. Similar results have been found for other tadpole species and in snails. For example, when exposed to cues of both giant water bugs and crayfish, snails preferentially responded to cues of water bugs, which is generally the more lethal predator [30, 33]. Similarly, *R. sylvatica* tadpoles have been shown to respond to the more lethal of two predators when exposed to mixtures of predator cues [40]. In instances where dragonfly larvae are a more dangerous tadpole predator than fish, tadpoles have responded to mixed predator cues with a phenotype more appropriate to the dragonfly nymph [39].

Our findings thus add another layer to our understanding of how prey have evolved in response to multiple predators that exert different selection pressures on the phenotype. Freshwater prey animals clearly have an ability to detect and integrate small amounts of predation cues from multiple sources into their developing phenotype [37, 38, 55, 59]. While multiple sensory modalities can be involved when prey detect predators [6], it appears that predation cues are primarily olfactory in nature, a combination of molecules that are released from injured prey animals themselves and those released from predators as they digest consumed prey [60, 61].

Clearly the cues that come from different predators are unique, or at least have enough unique molecular signatures to be perceived as distinct by prey. Tadpole olfaction occurs by olfactory receptor neurons embedded in the main olfactory epithelium, which project to the main olfactory bulb, at which point the scent is translated into a neural signal that can be interpreted by the brain [62, 63]. Scents that indicate predation are likely to trigger cascades of differentially expressed genes which result in changes to the phenotype, as has been seen following diet shifts in tadpoles of *Spea bombifrons* [64]. Assuming that different predators' scents do indeed trigger separate olfactory receptors, it is easy to imagine how exposure to multiple predation cues could result in the up- or down-regulation of separate genes associated with the response to each predator. That said, other aspects of the response to each predator likely involve differential expression of a single pathway. For example, in *D. ebraccatus* tail spot pigmentation increases dramatically with exposure to dragonflies and decreases with fish [19]. As such, it seems logical to assume that odors from each predator trigger opposing changes in the same gene or genes responsible for tail coloration. Our results here indicate that the smell of fish predation largely overrides any changes in gene expression that might result from smelling dragonfly predation.

Research with a number of different prey species—tadpoles, snails, freshwater crustaceans—has demonstrated that predator-induced phenotypic plasticity can be adaptive but that

being mismatched to a predator is disadvantageous [20, 21, 55, 59]. Our results here indicate that such mismatches may be relatively rare in nature. Prey species such as *D. ebraccatus* are highly sensitive to the cues of risk present in their environment and develop phenotypes according to that risk. Unless a particular habitat faces an unforeseen influx of a new predator, such as might occur after a flood or with a novel invasive species, prey animals appear likely to have phenotypes well-suited to the risks they face.

## Acknowledgments

Thank you to the Vassar College Animal Care staff for maintaining the research colony used in this research.

## Author Contributions

**Conceptualization:** Dean M. Rosenthal, Luana Deng, Justin C. Touchon.

**Data curation:** Dean M. Rosenthal, Justin C. Touchon.

**Formal analysis:** Dean M. Rosenthal, Justin C. Touchon.

**Funding acquisition:** Justin C. Touchon.

**Investigation:** Dean M. Rosenthal, Tarif Rose, Justin C. Touchon.

**Methodology:** Luana Deng, Justin C. Touchon.

**Writing – original draft:** Dean M. Rosenthal, Justin C. Touchon.

**Writing – review & editing:** Dean M. Rosenthal, Justin C. Touchon.

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
