## [Decision Letter · Decision Letter 0]

20 Feb 2023

PONE-D-23-01508

One of these things is not like the other: mixed predator cues result in lopsided phenotypic responses in a Neotropical tadpole

PLOS ONE

Dear Dr. Touchon,

Thank you for submitting your manuscript to PLOS ONE. After careful consideration, we have decided that your manuscript does not meet our criteria for publication and must therefore be rejected.

Specifically:

I have now received the comments from one reviewer and because the manuscript is within my field of expertise I have also read it in depth. Based on the reviewer’s comments and my own understanding, the experiments were not tested with a proper statistical design as they were unreplicated. Statistically, if the number of treatment groups are the same as the number of treatments there is only 1 replicate. In the method section you write:

*“In experiment one, we used 80 tadpoles from each family, set up in **eight groups** of 10 tadpoles each in 2 L of water. In experiment two, we used 76–88 tadpoles from each family, set up in** four sets** of approximately 20 tadpoles each in 4 L of water.”*

*“Experiment one contained four predator cue treatments (high, medium, low and no predator cues) crossed with two predators (mosquitofish, Gambusia affinis, or locally caught aeshnid dragonfly larvae) for** a total of eight treatments**.”*

*“Experiment two contained** four treatments:** no-predator control, fish, dragonfly, and 168 combined cues.”*

Unfortunately, this cannot be interpreted in another way as the experiments only had 1 replicate each and thus the statistical analyses are flawed. The idea behind and conceptual framing are very interesting but to consider a publication of the results would require fully replicated experiments.

I am sorry that we cannot be more positive on this occasion, but hope that you appreciate the reasons for this decision.

Kind regards,

Peter Eklöv

Academic Editor

PLOS ONE

Reviewers' comments:

Reviewer's Responses to Questions

**Comments to the Author**

1. Is the manuscript technically sound, and do the data support the conclusions?

Reviewer #1: No

2. Has the statistical analysis been performed appropriately and rigorously? 

Reviewer #1: Yes

3. Have the authors made all data underlying the findings in their manuscript fully available?

Reviewer #1: Yes

4. Is the manuscript presented in an intelligible fashion and written in standard English?

Reviewer #1: Yes

5. Review Comments to the Author

Reviewer #1: The MS presents interesting results from two experiments aimed at testing the fine-tuning of phenotypic responses of tadpoles to the amount of cue present from two types of predators, fish and dragonfly nymphs. Furhter, they show that the phenotypic responses to these two predators are opposite, and test what the plastic responses are like when cues from both predators are jointly presented. Nonetheless, from my understanding of the experimental design it looks like both experiments were unreplicated, which would be a fatal flaw in the design. In both experiments the number of groups of tadpoles was the same as the number of treatments, and it therefore follows that there was a single group of tadpoles, in a single container, assigned to each treatment.

The authors correctly include container as a random factor in their models to control for pseudoreplication, but that is no surrogate to actually implement a minimum level of experimental replication. Inclusion of container as a random factor in the mixed models would account for pseudoreplication, but it would not prevent increased type I error in the absence of replication, or even if true replication was small. It is otherwise a very nice study and the results fit perfectly the authors' expectations, so I would therefore encourage the authors to repeat the experiment with appropriate replication.

6. PLOS authors have the option to publish the peer review history of their article (what does this mean?). If published, this will include your full peer review and any attached files.

Reviewer #1: No

- - - - -

---

## [Author Response · Author response to Decision Letter 0]

6 Mar 2023

All comments except one from Reviewer 1 and from the Editor were positive, noting that the “conceptual framing [is] very interesting” and that it was “a very nice study.” The sole critique claimed that our study was unreplicated. This however is an incorrect interpretation of our methods. 

The section from our methods describing the replication of our experiments was as follows.

“Experiment one used eight unique families, whereas experiment two used six families. After hatching, tadpoles were pooled by family and then haphazardly divided into treatment groups. In experiment one, we used 80 tadpoles from each family, set up in eight groups of 10 tadpoles each in 2 L of water. In experiment two, we used 76–88 tadpoles from each family, set up in four sets of approximately 20 tadpoles each in 4 L of water. Extra hatchlings were used to feed predators in order to generate cues for the experiments.”

We apologize if the degree of replication was unclear. We thought that it would be clear given the number of animals being described in each experiment. For example, experiment one is described as having eight unique families, with 80 tadpoles used from each family, meaning 640 total tadpoles. If tadpoles were set up in groups of 10, that indicates 64 different units in the experiment, or eight full replicates of the eight treatments. The same logic applies to experiment two. However, in order to remove any confusion and prevent readers from similarly misinterpreting the design, we have modified the text of the methods as follows:

“Experiment one used eight unique families, whereas experiment two used six families. After hatching, tadpoles were pooled by family and then haphazardly divided into treatment groups. In experiment one, eight replicates of the experiment were setup (i.e., each family was a complete replicate of all treatments). We used 80 tadpoles from each family, set up in eight groups of 10 tadpoles each in 2 L of water. In experiment two, the experiment was once again replicated by family (i.e., all treatments were replicated six times). We used 76–88 tadpoles from each family, set up in four sets of approximately 20 tadpoles each in 4 L of water. Extra hatchlings were used to feed predators in order to generate cues for the experiments.”

---

## [Decision Letter · Decision Letter 1]

6 Apr 2023

PONE-D-23-01508R1One of these things is not like the other: mixed predator cues result in lopsided phenotypic responses in a Neotropical tadpolePLOS ONE

Dear Dr. Touchon,

Thank you for submitting your manuscript to PLOS ONE. After careful consideration, we feel that it has merit but does not fully meet PLOS ONE’s publication criteria as it currently stands. Therefore, we invite you to submit a revised version of the manuscript that addresses the points raised during the review process. Thanks for the clarifications of the experimental design that you included in the re-submitted manuscript. I have received responses from two reviewers and you can find minor comments from one of them below. Please, revise your text accordingly, as a response to this reviewer's comments, and in your resubmitted version of the manuscript include how you dealt with this reviewer's comments.

We look forward to receiving your revised manuscript.

Kind regards,

Peter Eklöv

Academic Editor

PLOS ONE

Journal Requirements:

"Funding was provided by Vassar College and the Vassar College Diving Into Research program."

"The author(s) received no specific funding for this work". 

Please include your amended statements within your cover letter; we will change the online submission form on your behalf. 5. Please include your full ethics statement in the ‘Methods’ section of your manuscript file. In your statement, please include the full name of the IRB or ethics committee who approved or waived your study, as well as whether or not you obtained informed written or verbal consent. If consent was waived for your study, please include this information in your statement as well.

Additional Editor Comments (if provided):

Reviewers' comments:

Reviewer's Responses to Questions

**Comments to the Author**

1. If the authors have adequately addressed your comments raised in a previous round of review and you feel that this manuscript is now acceptable for publication, you may indicate that here to bypass the “Comments to the Author” section, enter your conflict of interest statement in the “Confidential to Editor” section, and submit your "Accept" recommendation.

Reviewer #1: All comments have been addressed

Reviewer #2: (No Response)

2. Is the manuscript technically sound, and do the data support the conclusions?

Reviewer #1: Yes

Reviewer #2: Yes

3. Has the statistical analysis been performed appropriately and rigorously? 

Reviewer #1: Yes

Reviewer #2: Yes

4. Have the authors made all data underlying the findings in their manuscript fully available?

Reviewer #1: Yes

Reviewer #2: Yes

5. Is the manuscript presented in an intelligible fashion and written in standard English?

Reviewer #1: Yes

Reviewer #2: Yes

6. Review Comments to the Author

Reviewer #1: Thank you for clarifying the experimental design, and apologies for having erroneously interpreted that there was lack of replication in the study.

Reviewer #2: Review PONE-D-23-01508R1, Rosenthal et al.

The authors present an interesting study on effects of two predators (odonate larvae and fish), with contrasting selectivities, on tadpole phenotypes. They show that higher predator concentrations induce stronger defenses and that mixed predator cues lead to a phenotype seemingly directed against fish predation. They argue that this is a proof that defenses are directed against the stronger predator. Obviously, the manuscript had been reviewed before, as there were responses to the reviewers included.

The paper is interesting, experiments are generally well designed, methods adequate, and well written. I have just a few minor comments, which can be included easily.

General comment:

1. The authors cannot claim that “Fish are more lethal than dragonfly larvae and thus tadpoles responded most strongly to the more dangerous predator” (L. 32 and later), because this has not been shown. The kairomones are unknown and thus the concentration is not measurable. The same number of fish and odonate larvae have been used as predators but there is no reason to assume that the kairomone concentrations are equal (L. 36 ,”…they produce a stronger response to a more lethal predator even when the strength of cues is equal”). I accept that the results indicate that the tadpoles respond to the stronger predator. So I suggest to tune it down a bit.

Minor comment:

2. L. 55: The “phenotype inducing cues” additionally may even be a combination with prey density cues from conspecifics. Many prey species estimate their predation risk by integrating predator and prey concentrations (Tollrian et al. 2015, Scientific Reports 5, 1-9).

End of review

7. PLOS authors have the option to publish the peer review history of their article (what does this mean?). If published, this will include your full peer review and any attached files.

Reviewer #1: No

Reviewer #2: No

---

## [Author Response · Author response to Decision Letter 1]

10 Apr 2023

Below, you will find our detailed Response to Referees. We have uploaded a manuscript with tracked changes for your review.

Editor comments

We have updated our manuscript and figure files to follow the naming and style guidelines of PLOS One, including updating the author affiliations, and font sizes of level 1 and 2 headings. We have updated the Acknowledgements section to remove the funding statement and IACUC approval statement. The IACUC approval statement with approved protocol numbers has been placed at the beginning of the Materials and methods section. Our full funding disclosure is as follows:

Funding was provided by Vassar College and the Vassar College Diving Into Research program. Justin C. Touchon receives a salary from Vassar College. The funders had no role in study design, data collection and analysis, decision to publish, or preparation of the manuscript.

Note that there are no grant numbers to report with regards to the funding.

Response to Reviewer 2

Reviewer 2 made two comments, asking us to 1) soften our language in the Abstract about the responses of the tadpoles to mixed cues of our two predators and 2) to consider an addition reference in our Introduction. We have implemented both of these changes, which are shown below with new text in green.

Abstract, lines 42–46: “Previous studies have shown that fish are more lethal than dragonfly larvae; thus tadpoles responded most strongly to the more dangerous predator, even though the number of prey consumed by each predator was the same. This may be due to D. ebraccatus having evolved a stronger response to fish or because fish produce more kairomones than do dragonflies for a given amount of food.”

Introduction, lines 70–74: “Depending on the predator-prey relationship, the phenotype-inducing cue may be a combination of kairomones released from the predator alone, alarm pheromones produced by conspecific prey under attack, density based cues of conspecifics, and/or compounds and tissues derived from the mastication and digestion that occur with predation events [9-12].”

---

## [Editor Report · Decision Letter 2]

7 May 2023

One of these things is not like the other: mixed predator cues result in lopsided phenotypic responses in a Neotropical tadpole

PONE-D-23-01508R2

Dear Dr. Touchon,

We’re pleased to inform you that your manuscript has been judged scientifically suitable for publication and will be formally accepted for publication once it meets all outstanding technical requirements.

Kind regards,

Peter Eklöv

Academic Editor

PLOS ONE
---

## [Editor Report · Acceptance letter]

15 May 2023

PONE-D-23-01508R2 

One of these things is not like the other: mixed predator cues result in lopsided phenotypic responses in a Neotropical tadpole 

Dear Dr. Touchon:

I'm pleased to inform you that your manuscript has been deemed suitable for publication in PLOS ONE. Congratulations! Your manuscript is now with our production department. 

Kind regards, 

on behalf of

Dr. Peter Eklöv 

Academic Editor

PLOS ONE